# Field Modeling of the Influence of Defects Caused by Bending of Conductive Textronic Layers on Their Electrical Conductivity

**DOI:** 10.3390/s23031487

**Published:** 2023-01-29

**Authors:** Stanisław Pawłowski, Jolanta Plewako, Ewa Korzeniewska

**Affiliations:** 1Department of Electrodynamics and Electrical Machine Systems, Faculty of Electrical and Computer Engineering, Rzeszow University of Technology, al. Powstańców Warszawy 12, 35-959 Rzeszow, Poland; 2Department of Power Electronics and Power Engineering, Faculty of Electrical and Computer Engineering, Rzeszow University of Technology, al. Powstańców Warszawy 12, 35-959 Rzeszow, Poland; 3Institute of Electrical Engineering Systems, Lodz University of Technology, Stefanowskiego 18 Street, 90-537 Lodz, Poland

**Keywords:** thin films, wearable electronics, textronics, cracks, modeling of electroconductivity phenomena, electroconductivity

## Abstract

One of the critical parameters of thin-film electrically conductive structures in wearable electronics systems is their conductivity. In the process of using such structures, especially during bending, defects and microcracks appear that affect their electrical parameters. Understanding these phenomena in the case of thin layers made on flexible substrates, including textile ones, which are incorporated in sensors that monitor vital functions, is a key aspect when applying such solutions. Cracks and defects in such structures appearing during their use may be critical for the correct operation of such systems. In this study, the influence of defects resulting from the repeated bending of the conductive layer on its conductivity is analyzed. The anisotropic and partly stochastic characteristics of the defects are also taken into account. The defects are modeled in the form of broken lines, whose segments are generated in successive iterative steps, thus simulating the gradual wear of the layer material. The lengths and inclinations of these sections are determined randomly, which makes it possible to consider the irregularity of real defects of this type. It was found that near the percolation threshold, defects with a more irregular shape have a dominant effect on the reduction of conductivity due to the greater probability of their connection. The simulation results were compared with the experimental data. It was found that the dependence of the conductivity of the conductive layer on the number of bends is logarithmic. This allowed for the derivation of a formula linking the iteration number of the simulation procedure with the number of bends. Improving the strength of such layers is a technological challenge for researchers.

## 1. Introduction

The development of modern technologies sparks innovative solutions in many engineering areas, including wearable electronics. The miniaturization of electronic components and social expectations regarding the permanent availability of many technical devices determine the development of textronics, i.e., a field combining textiles, electronics, and computer science. The production of electronic systems with sensors on a textile substrate is also a challenge for materials engineering. Placing sensors and electroconductive structures on fabrics, including clothing, makes the clothing not only protective but also provides the clothing with previously unattainable functionality. It is the added value that gains supporters in many applications in fields such as the military, workwear, fashion, or care for the elderly or those requiring medical supervision [1,2,3,4,5,6,7]. Equipping fabrics with elements of wearable electronics requires the production of structures with the desired electrical properties, including structures with a high conductivity value. Such structures should also be flexible and resistant to mechanical damage, particularly stretching, bending, and abrasion. Representative examples of biosensors created as wearable electronic on various non-textile substrates are described in the review by Kim’s group [8].

The weaving of metal wires or conductive threads [1], or covering them with metal [9] or conductive polymers [10] in the process of knitting [11] or weaving, are among the known methods of producing such elements. The weaving of threads reduces the elasticity of textile products and thus affects their wearability-related comfort. The production of electrically conductive structures in this technology requires the design of systems at the initial stage of fabric production. Other technologies that enable the production of electrically conductive structures on existing materials include embroidery [12], screen printing [13], inkjet printing [14,15], electroless coating [16], electroconductive coating [17], chemical vacuum deposition (CVD) [18], sputtering [19,20,21], and physical vacuum deposition (PVD) [22,23,24]. Physical vacuum deposition methods include cathodic arc deposition, sputtering, ion plating [25,26], or thermal deposition [23]. This technology is used in the application of coatings of significant technological importance in mechanical processing, among other fields. Coatings obtained by the PVD method are characterized by increased hardness. In addition, they have increased resistance to damage, such as abrasion, breakage, or chipping. Due to the known properties of such coatings, this technology has also found applications in textronics.

The PVD technique uses various physical phenomena, such as sputtering, sublimation (evaporation), or sputtering, which are intensified by additional reactive processes (using reactive gases that contribute to the formation of a layer of high hardness), activation (by activating the ionization processes of gases and vapors metals), or both [22,23,24]. The connection at the interface between the coating and the surface of the tool is adhesive (rarely, adhesive-diffusion), and its strength is largely determined by the cleanliness of the coated surface. Therefore, it is necessary to carry out a cleaning process before applying the coating. Physical vapor deposition is the formation of a thin layer on a substrate as a result of the deposition of ions, atoms, or molecules in a high vacuum [27]. In the process of thermal deposition, a source of vapor (deposited metal) is heated and placed inside a boat made of metal with a higher melting point, e.g., tungsten. The process takes place in a vacuum of about 10^−5^ Pa. As a result of the supplied energy, the metal placed in the tungsten boat melts, transforms into a gaseous state and, as a result of movement inside the vacuum chamber, settles on a substrate with a much lower temperature. The formation of the coating takes place on a specific substrate as a result of vapor condensation and coating growth. 

In addition, during the use of the produced structures, additional damage and cracks appear, which are of particular importance in the case of sensory structures, where even small changes can be significant in detecting changes in the processed signals. 

A description of the changes in the electrical properties of textronic structures produced on composite textile substrates has been described in [28,29,30]. As a result of the formation of microcracks, changes to both the conductivity of the created structure and the amount of locally generated heat appear [31].

The main aim of this work is to investigate, via numerical simulations, the influence of cracks in the conductive textronic layers on their conductive properties caused by their repeated bending. This paper proposes a numerical procedure for generating lines that model this type of crack in subsequent iterative steps, enabling the simulation of their formation and growth in the process of the gradual wear of the layer material. Based on the method of integral equations, a numerical program was created to calculate the distribution of the current flow field and the electrical conductivity of the defective layer. By accounting for the comparison of the simulation results with the results of physical experiments, a relationship was derived that allows for the linking of the number of iterations of the procedure with the number of bends of the textronic conductive layer. 

Due to the developing nature of this technology, i.e., textronics, the presented results constitute a contribution to understanding the phenomena occurring in thin layers produced on textile substrates appearing in the process of using such structures.

## 2. Materials and Methods

### 2.1. Experiment

Thin electroconductive layers produced in the vacuum deposition process according to the procedure described in the authors’ earlier article [6] were subjected to cyclic bending. The thin-film structures were placed in clamps and then bent by 45° both upwards and downwards, which amounts to 90° of deformation in total. The entire process of mechanical bending was described in the previous paper [28], in which the authors analyzed the dependence of the resistance of the metallic layer formed on a flexible substrate on the number of bends. In the experiment, the resistance of conductive paths was measured using the two-electrode method via the Picotest M3500A ohmmeter. A microscopic photo of a thin, silver, metallic layer with a thickness of 250 nm produced in the experiment is shown in Figure 1. The photo was taken using SEM Hitachi S-4200 scanning microscope (Tokyo, Japan).

Due to the stochastic nature of the production of the structure in the PVD process, surface and structural defects arise that affect the electrical parameters of the produced layer (Figure 1).

### 2.2. Description of the Model and Formulation of the Problem

The small transverse dimensions of the conductive thin layers in relation to their length is one of the characteristic features of defects resulting from repeated bending. In the described model, it was assumed that they can be treated as infinitely thin broken lines with the properties of an ideal dielectric, i.e., constituting an impenetrable barrier for electric current (Figure 2). Moreover, it is assumed that:(1)The conductive layer is infinitely thin and rectangular in shape;(2)The conductivity of the layer is constant;(3)The surrounding layer is an ideal dielectric;(4)There is a constant electric voltage between two parallel edges of the layer;(5)The stationary operating state of the system can be analyzed (electromagnetic field does not depend on time);(6)There are no unbalanced electric charges in the system;(7)Quantum effects are omitted (e.g., tunneling of current carriers through the defect areas).

Herein, the analysis of this phenomenon concerns only the thin layer formed on the substrate. The mechanical strength and geometric dimensions of the substrate were not taken into account.

The thickness of the conductive layers obtained in the PVD process does not exceed one micrometer, which, in comparison, to their other dimensions (in the order of millimeters) justifies treating them as infinitely thin. This assumption makes it possible to reduce the problem of calculating the flow field of the current density vector to a two-dimensional problem.

Due to the small thickness of the analyzed layers and their defects, a question about the legitimacy of ignoring quantum effects may arise. Quantum effects such as electrical conduction by tunneling or quantum point contacts play an important role at the atomic scale. They can still be significant in the range of nanometers. Defect dimensions of larger quantum effects disappear more quickly. For example, $2mVa^{2}/\hbar^{2}$ is the measure of opacity of a potential barrier of width *a* and height *V* that has a decisive influence on the probability of tunneling [32].

Typical thicknesses of the defects of conductive layers, deposited with the PVD technique analyzed in the paper, are of the order of at least hundreds of nanometers, and the height of the potential barrier at the metal–dielectric interface is of the order of several electron volts. In this case, the opacity of the barrier is about 10^6^ >> 1. Thus, the tunneling effect is completely negligible [33].

Under the mentioned assumptions, the electric field is a potential field:(1)E=−gradϕ
where *φ* is the scalar electric potential corresponding the Laplace equation:(2)Δϕ=0

According to Ohm’s local law, the vector of current density ***J*** can be directly related to the electric field strength
(3)J=γE
where *γ* denotes layer conductivity.

At the L_1_ and L_2_ edges of the layer (see Figure 1), there are:(4)ϕ(0,y)=u,ϕ(l,y)=0

The normal components of the vector ***J*** on the edges L_3_ and L_4_, as well as on the defect lines Γ*_k_*, are zero, which, according to (1) and (3), means that the potential on L_3_ and L_4_ satisfies the Neumann-type conditions:(5)$${\frac{\partial \phi }{\partial n}|}_{y=0}={\frac{\partial \phi }{\partial n}|}_{y=H}={\frac{\partial \phi }{\partial n}|}_{\Gamma_{k}}=0,k=1,\ldots ,N$$

The problem is to find a solution to the Laplace Equation (2) with mixed boundary conditions (4,5). To solve it, the method of integral equations was used. Its detailed description and algorithm are presented in [34,35]. 

The problem of the cracking and stretching of thin-film structures in wearable devices was discussed in an article by Zhang et al. [36]. They proposed a rolling strategy to prepare conductive fibers with high stretchability based on a spiral structure. In their work, the group used the commercial software ANSYS and finite-element-modeling (FEM) simulation to ascertain the stress distribution of cracks.

Lo et al. [37] also used the finite element simulation method in the COMSOL package to analyze micro-cracks in thin electro-conductive structures to understand the relationship between micro-cracks and tensile strains. In addition, Kwon’s group [38] used the same package and simulation method to analyze the distribution of stress in the conductive fibers as the phenomena occurring in the elements of wearable electronics. 

The simulations in the presented paper are executed using proprietary software employing the integral equation method, which is an alternative to the finite element method and has many comparative advantages:It does not require discretization of the interior of the considered area, only its edges, resulting in a significantly smaller numerical model of the problem;The solution is presented in an analytical form that exactly satisfies Laplace’s Equation (2);Its numerical error concerns only the fulfillment of boundary conditions;It enables a relatively simple evaluation of the error of the solution.

The numerical application was implemented in the Microsoft Visual Studio 2010 environment using Fortran 77. 

The correctness of the adopted assumptions and the applied calculation method was verified empirically in [34].

### 2.3. Defect Line Generation Procedure

The characteristic features of cracks in thin conductive layers resulting from repeated bending are their small transverse dimensions in relation to their length, distinguished directionality (anisotropy), and stochastic character, both in terms of their arrangement and shape. The purpose of the described procedure is to numerically generate the coordinates of the polyline vertices modeling such defects and to simulate the process of their growth and the formation of new ones in the process of wearing of the conductive path material. 

The procedure’s input parameters are: the dimensions of the rectangular area of the conductive path in which the defect lines will be generated —$x\in \left[x_{\min},x_{\max}\right]$, $y\in \left[y_{max},\;\;y_{min}\right]$ the average distance between successive vertices of the line *d*_0_; the standard deviation of this value *σ_d_*; the average angle of inclination of the line segments with respect to the axis of the path (direction of the applied electric field) *α*_0_; the standard deviation of this angle *σ_α_*; and the starting number for the random number generator *R_n_*, which enables the acquirement of various random realizations of the defect line.

The procedure is iterative, which corresponds to the bending cycle in the process of wearing electrically conductive structures used, among other applications, in thin-film sensors, which can be applied in wearable electronics systems. Its subsequent steps can be treated as stages of wear of the conductive path material consisting of the formation of new cracks and their growth. In the first step, the starting point $P_{0}^{1}\;$ of the first defect line is selected (the upper indicator denotes the number of lines, the lower one denotes the successive vertices of a given line), whose coordinates are determined using a random number generator. Then, two successive points of this line are determined: $P_{1}^{1}$, $P_{-1}^{1}$. Their locations are determined by vectors $\vec{P_{0}^{1}P_{1}^{1}}$, $\vec{P_{0}^{1}P_{-1}^{1}}$, whose lengths are determined randomly according to Gaussian distribution with mean *d*_0_ and standard deviation *σ_d_*. Similarly, the inclination angles *α*_1_, *α*_−1_ of these vectors to the path axis are determined randomly according to the Gaussian distribution with mean *α*_0_ and *α*_0_ + π (respectively) and standard deviation *σ_α_*. The points generated in this way, $P_{-1}^{1}$, $P_{0}^{1}$, and $P_{1}^{1}$, constitute three successive vertices’ first defect lines. In the second iteration step, the coordinates of the points $P_{-1}^{2}$, $P_{0}^{2}$, and $P_{1}^{2}$ of the second line and subsequent points $P_{2}^{1}$ and $P_{-2}^{1}$ of the first line are generated in a similar way. The successive iterative steps are analogous to those described above.

The addition of two vertices to each defect line in the subsequent iteration steps with the average angle of inclination of the vectors $\vec{P_{k}^{n}P_{k+1}^{n}}$, $\vec{P_{-k}^{n}P_{-k-1}^{n}}$, differing by the value of π, is performed to ensure that these lines grow at both their ends independently, but with the same probability of deviation of the length and angle of inclination of their subsequent sections.

As a result, after *N* iteration steps, *N* series of points are generated, constituting vertices of broken lines of various lengths and shapes. With a given angle *α*_0_ and a small dispersion *σ_α_*, the obtained lines show clear directionality (in the borderline case, *σ_α_* = 0, they will be parallel segments of a straight line), as well as real defects resulting from repeated bending of the conductive path (see Figure 3a,b). As the parameter *σ_α_* increases, the image of the line gradually loses its anisotropic character and becomes more chaotic (see Figure 3c,d).

## 3. Results

### 3.1. Results of Simulations

To examine the dependence of the influence of the process of the formation and growth of defects of the conductive layer resulting from repeated bending on its conductivity, a number of numerical simulations were performed based on the described model. Some of the results are presented in this paper. They concern the following parameters of the analyzed model:$$x\in \left[0,\;10\right]\;,\;y\in \left[0,\;10\right],\;u=1\;V,\;d_{0}=0.05÷\;0.7,\;\sigma_{d}=0.01÷\;0.24,\alpha_{0}=90^{\circ },\;\mathrm{and}\;\sigma_{\alpha }=5^{\circ }÷{\;45}^{\circ }$$

(units of size with the dimension of length are conventional; their absolute values are not significant for the results presented below).

Figure 4 presents examples of the simulations of current flow field distributions for the conductive layer model with defect lines generated at *d*_0_ = 0.2, *σ_d_* = 0.04, $\sigma_{\alpha }=25^{\circ }$ and the various stages of iteration.

The plots in Figure 5 show the calculated current flow field distributions for *d*_0_ = 0.2, *σ_d_* = 0.04, $\sigma_{\alpha }=5^{\circ },\;25^{\circ }{,\;45}^{\circ }$ after 13 iteration steps.

Figure 6 shows the graphs of the calculated dependences of the electrical conductivity (G) of the conductive layer with defects in relation to the conductivity (*G*_0_) of the layer without defects in successive iteration steps and for various random implementations (differently set parameters *R_n_*) at $\sigma_{\alpha }=25^{\circ }$. As expected, the randomness of the distribution and shape of the defect lines do not significantly affect the conductivity of the layer in the first few steps of the iteration, when the number of defects and their lengths are relatively small. As the number of defects and their length increases, the scattering of the calculated conductivities increases, reaching the highest values near the point at which the current flow is interrupted (percolation threshold). Changes in the conductivity of thin electroconductive layers associated with the appearance of cracks and defects described in this article, under the influence of bending in a physical experiment, were described in the previous work by the authors [31].

The following graphs concern the study of the dependence of the electrical conductivity of the conductive layer on the parameters of the defect lines used in the simulation. The graph in Figure 7 shows the dependence of the *G*/*G*_0_ ratio on the ratio of the average distance between the vertices of the defect line *d*_0_ to the layer width *y_max_* at *σ_d_* = 0.2*d*_0_. $\sigma_{\alpha }=25^{\circ }$, after 10 iteration steps.

The graphs in Figure 8 show changes in the conductivity of the conductive layer in subsequent iteration steps for variously set parameters of *σ_α_*.

### 3.2. Comparison to the Experiment

Examples of the results of the described experiments are presented in the graph in Figure 9a. This graph concerns the measured values of the conductivity of a silver layer applied with the PVD technique on Cordura as a function of the number of bends. The plot in Figure 9b shows the same results on a logarithmic scale on the abscissa axis. As can be seen, the measurement points are arranged close to a straight line, which suggests a logarithmic dependence of the electrical conductivity on the number of bends. The red line marks the graph of the function matching the measurement results of the form:(6)f(n)=a−bln(n+c)

The box placed in Figure 9b shows the values of parameters *a*, *b*, and *c* and the statistical parameters determining the degree of matching, calculated using the OriginPro7 program (at the confidence level of 0.95).

Figure 10 shows that the dependence of G/G_0_ on the number of iterations of the described procedure for several initial steps is of a different nature, but it appears that subsequent iteration steps can be related to the number of bends, thus obtaining a satisfactory agreement between the simulation and measurement results.

Figure 10 shows the fitting of the central part of the graph of the selected simulation (see Figure 6) with a linear function:(7)g(N)=A+BN

By comparing (6) and (7), the following formula is obtained
(8)$$n=Ce^{-pN}-c$$
where
(9)$$C=e^{\frac{a-A}{b}},\;p=\frac{B}{b}$$

Relationship (8) allows for the assignment of the corresponding number of bends n to the iteration step *N* of the simulation procedure. Specific values of the parameters *a*, *b*, and *c* depend on the material properties of the conductive layer and the type of textile substrate. 

The constants *A*, *B*, and *C* depend on the specific random implementation of the simulation. However, based on a series of simulations, it was found that this does not have a significant impact on the consistency of the simulation results with respect to the experimental data.

Figure 11 shows a comparison of the experimental results with the iteration steps in accidence with (8) in the simulation procedure for conductive layers of Ag (Figure 11a) and Au (Figure 11b) on Cordura.

## 4. Discussion and Conclusions

The obtained simulation results regarding the mapping of structural cracks are consistent with the defects observed in the microscopic photo presented in Figure 1. 

As expected, the randomness of the distribution and shape of the defect lines do not significantly affect the conductivity of the layer in the first few steps of the iteration, at which point the number of defects and their lengths are relatively small. As the number of defects and their length increases, the scattering of the calculated conductivities increases, reaching the highest values near the point at which the current flow is interrupted (the percolation threshold). 

As expected, the randomness of the distribution and shape of the defect lines do not significantly affect the conductivity of the layer in the first few steps of the iteration, at which point the number of defects and their lengths are relatively small (Figure 4 and Figure 6). As the number of defects and their length increase, along with the decrease in the conductivity of the layer, its dispersion increases, reaching the highest values near the percolation threshold, i.e., the point at which the current flow is interrupted.

The number of iterations after which the percolation threshold is reached depends on many factors, namely, the size of the examined area and all parameters of the defect line (*d*_0_, *σ_d_*, *α*_0_, and *σ_α_*). The graph in Figure 5 shows that even if these parameters are constant, the number of iterations after which the layer ceases to conduct current may differ significantly for different random implementations of the simulation.

The impact of the parameters *d*_0_ and *σ_d_* on the layer conductivity in terms of quality is rather obvious: their increase causes the defect lines to grow faster in the iterative process, which results in a faster decrease in the layer conductivity (see Figure 7).

The influence of the value of parameter *σ_α_* on the conductivity of the layer turns out to be more complex. With the angle *α*_0_ = 90^o^ assumed in the described simulations, increasing the value of *σ_α_* causes a statistical decrease in the size of defects in the direction perpendicular to the applied electric field (Figure 3 and Figure 5), which should generally facilitate the flow of electric current and increase the effective conductivity of the layer. From the graphs in Figure 8 it can be seen that this occurs up to the 17th step of the iteration. However, as the percolation threshold becomes closer, this rule usually ceases to apply, which is clearly visible in the location of the percolation threshold for different values of *σ_α_*. In the range of *σ_α_* from 5° to 45°, there is a clear shift of the percolation threshold towards smaller values of *N*. This effect should be explained by the fact that a higher value of *σ_α_* increases the probability of joining different defect lines, which may lead to the effective elongation of fault lines and the faster interruption of the current flow path. With a further increase in *σ_α_* above 45°, the percolation threshold shifts back towards higher values; thus, in this range, the effect of decreasing the size of defects in the direction perpendicular to the direction of the field forcing the current flow begins to dominate.

Based on a series of experiments, it was found that the dependence of the conductivity of a thin conductive layer on the number of bends is logarithmic and can be accurately described by relationship (5). Assuming a linear adjustment of the dependence of conductivity on the number of simulation procedure iterations in the central part of the characteristics (see Figure 10), Formula (8) was obtained. This enabled the matching of the number of iterations of the simulation procedure with the number of bends. The results of the comparison of numerical calculations using Formula (8) with the experimental data for the Ag and Au layers on Cordura indicate a very good agreement between the simulations and experimental results (Figure 10), which confirms the correctness of the assumptions of the model and calculation method.

## Figures and Tables

**Figure 1 sensors-23-01487-f001:**
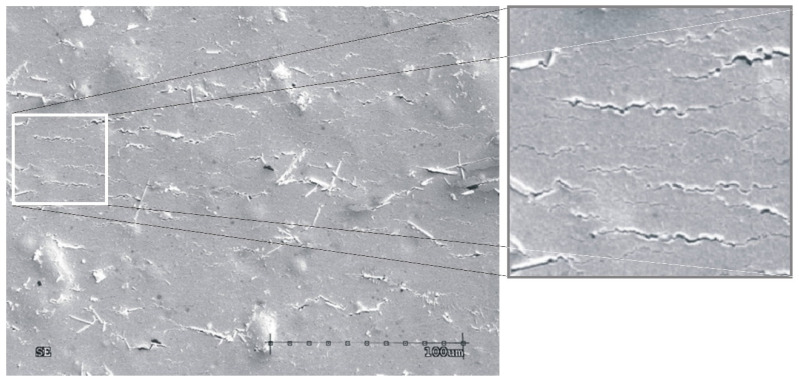
The microscopic image of created metallic layer under physical vacuum deposition (the dimensions of the enlarged box 150 × 150 µm).

**Figure 2 sensors-23-01487-f002:**
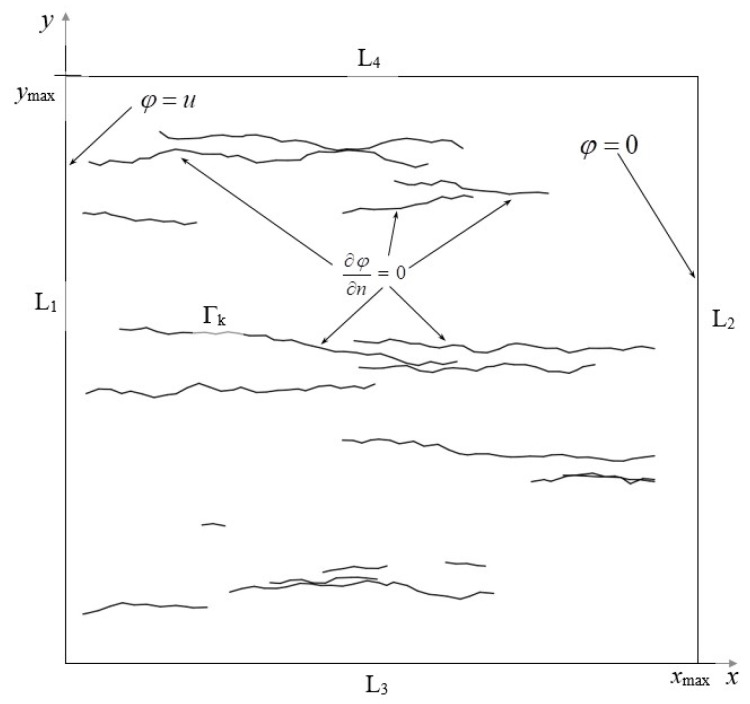
The analyzed model of a conductive layer with line defects.

**Figure 3 sensors-23-01487-f003:**
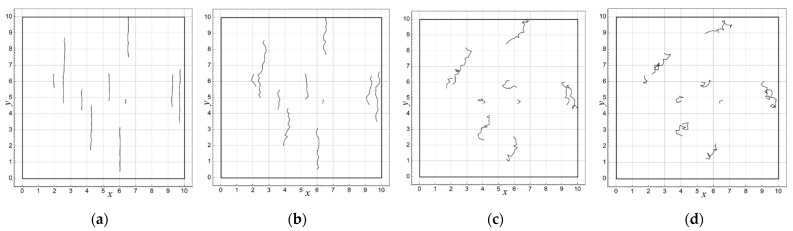
Examples of generated linear defects according to the described procedure after ten iterative steps, for *α*_0_ = 90° and different parameter values *σ_α_*: (**a**) 5°, (**b**) 25°, (**c**) 45°, and (**d**) 90°.

**Figure 4 sensors-23-01487-f004:**
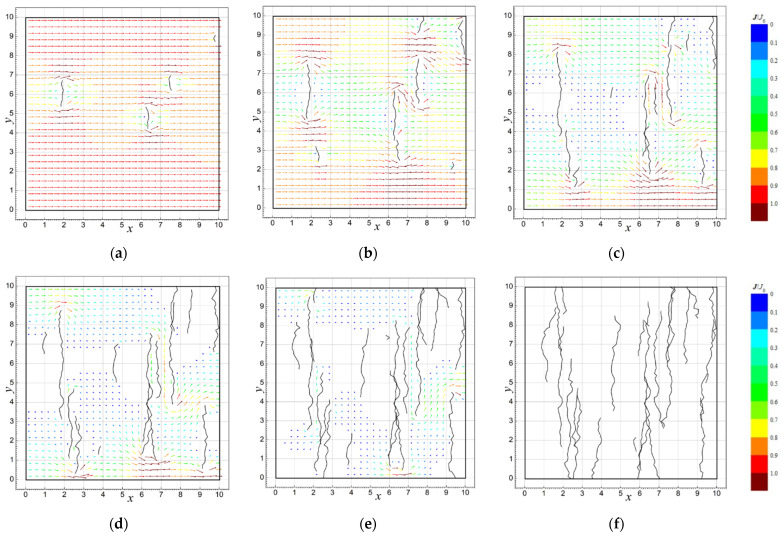
The simulation results of the relative current flow field *J*/*J*_0_ in the analyzed model of the conductive layer with line defects after (**a**) 4, (**b**) 8, (**c**) 12, (**d**) 16, (**e**) 20, and (**f**) 24 iteration steps; *d*_0_ = 0.2, *σ_d_* = 0.04, and σ*_α_* = 25*°*.

**Figure 5 sensors-23-01487-f005:**
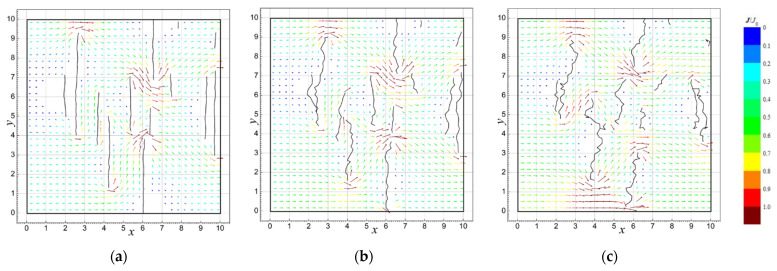
The results of simulation the relative current flow field *J*/*J*_0_ in the analyzed model of the conductive layer with line defects for different values of the parameter *σ_α_*: (**a**) 5°, (**b**) 25°, and (**c**) 45°; *N* = 13 (units of *x*, *y* are conventional).

**Figure 6 sensors-23-01487-f006:**
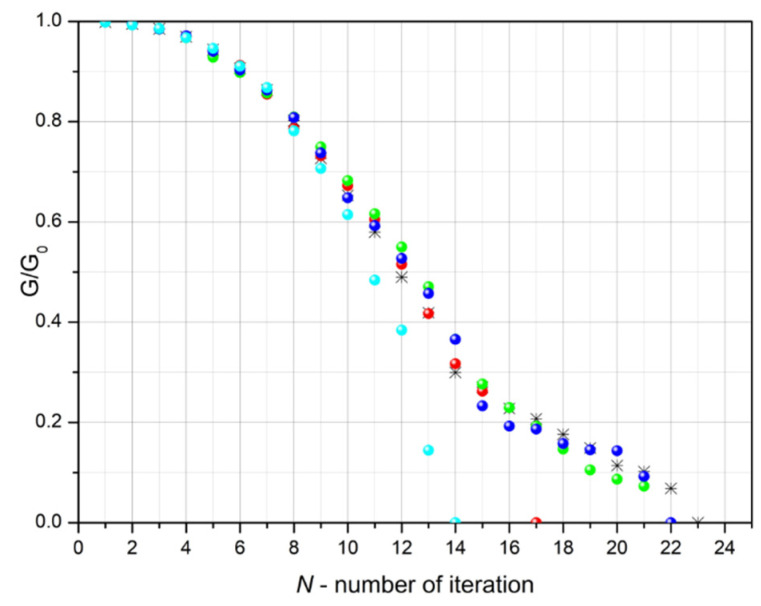
The conductivity ratio of the conductive layer with defects to the layer without defects calculated in successive iteration steps for various random realizations (various colors) of defects (*d*_0_ = 0.2, *σ_d_* = 0.04, $\sigma_{\alpha }=25^{\circ }$).

**Figure 7 sensors-23-01487-f007:**
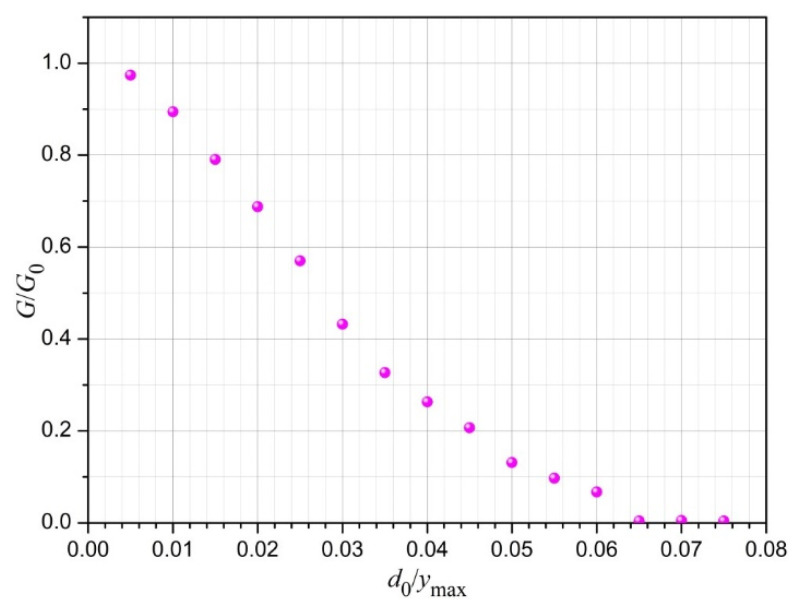
The ratio of the conductivity of the conductive layer with defects to the conductivity of the layer without defects depending on the ratio of the average length of the defect line segments *d_0_* in relation to the layer width *y_max_*. Other simulation parameters are as follows: *σ_d_* = 0.2 *d*_0_, $\sigma_{\alpha }=25^{\circ }$, and *N* = 10.

**Figure 8 sensors-23-01487-f008:**
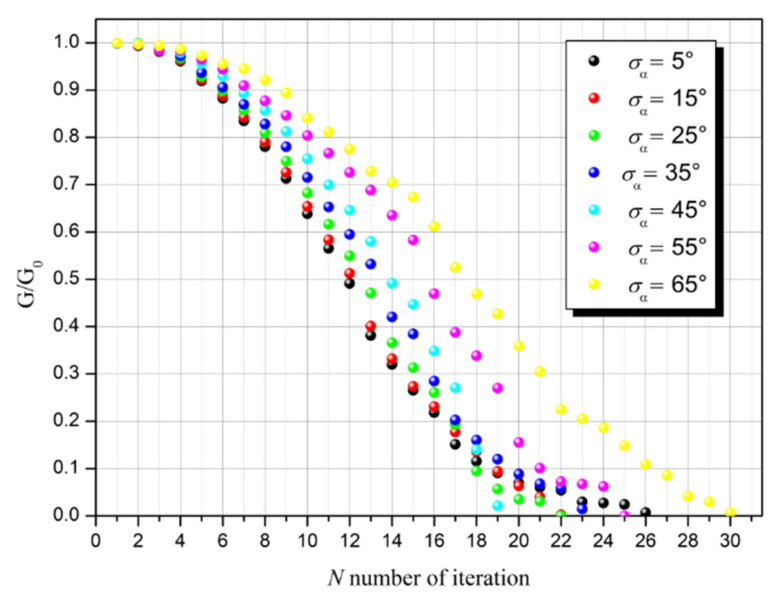
The conductivity ratio of the conductive layer with defects to the layer without defects calculated in subsequent iteration steps for various parameters of *σ_α_*.

**Figure 9 sensors-23-01487-f009:**
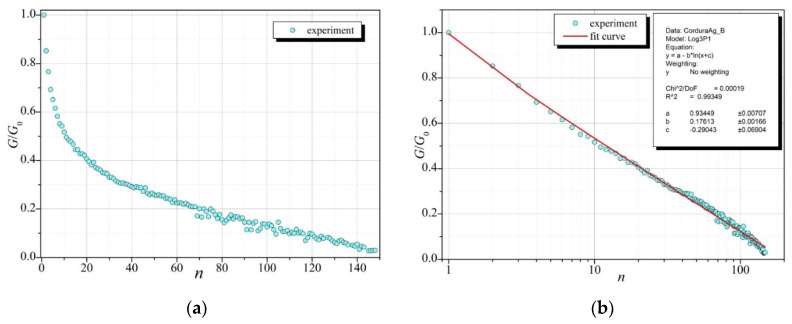
The ratio of conductivity of the silver layer (Ag on Cordura) with defects to the conductivity of the sample without defects depending on the number of bends with respect to (**a**) the results of the experiment; (**b**) fitting with a logarithmic function (6). The inset in the Figure 9b shows the values of parameters *a*, *b*, and *c* and statistical parameters (Chi^2/^DoF—Test Chi^2^/Degrees of Freedom, R^2^—coefficient of determination) determining the degree of matching, calculated using the OriginPro7 program (at the confidence level of 0.95).

**Figure 10 sensors-23-01487-f010:**
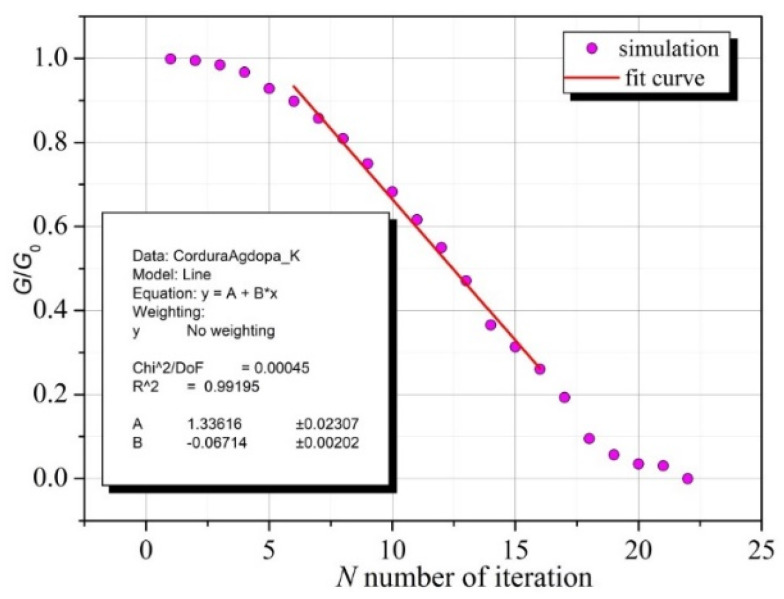
Fitting part of the results of the selected simulation (see Figure 6) with a linear function: *g*(*N*) = *A* + *BN*. The inset presents the values of factors calculated during fitting the simulation and experimental data.

**Figure 11 sensors-23-01487-f011:**
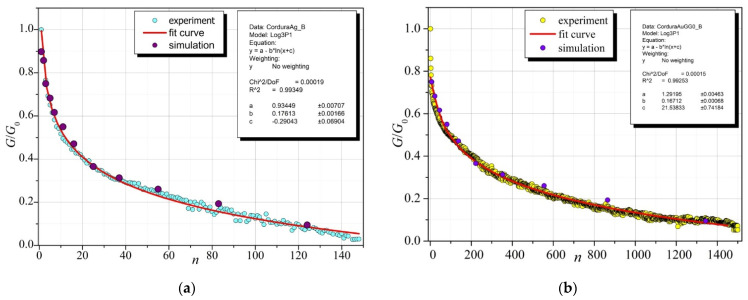
Comparison of experimental results with simulation results for two conductive layers on Cordura: (**a**) for Ag; (**b**) for Au. Figure 11a,b: the inset shows the values of parameters a, b, and c and statistical parameters (Chi^2^/DoF—Test Chi^2^/Degrees of Freedom; R^2^—coefficient of determination) determining the degree of matching, calculated using the OriginPro7 program (at the confidence level of 0.95).

## Data Availability

Not applicable.

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
