# Peer review of "Field Modeling of the Influence of Defects Caused by Bending of Conductive Textronic Layers on Their Electrical Conductivity"

_sensors, 2023, doi:10.3390/s23031487_

Round 1

Reviewer 1 Report (Previous Reviewer 2)

Dear authors

I appreciate the effort placed in individually replying the first round of comments. After reading the resubmitted version, I must maintain my decision to not accept the manuscript. In order to support my decision, I copy below some fragments of your manuscript, followed by a discussion:

From the abstract: “It was found that near the percolation threshold, defects with more irregular shapes have a dominant effect on conductivity reduction due to the greater probability of their connection.”

As mentioned in the abstract, percolation regime was considered as the start point for the crack simulation. When cracks occur, percolation is interrupted and quantum tunneling dominates if particle separation remains within a few nanometers; this is a phenomenon observed on 2D and 3D materials, which has been thoroughly studied in condensed matter physics [1, 2].

Therefore, it is not possible to conduct a crack simulation by relying only on classical physics, see equations (1) – (5) in the resubmitted manuscript.

References

[1] doi: https://doi.org/10.1016/0921-4526(95)00602-8

[2] doi: https://doi.org/10.1103/PhysRevB.59.3168

Author Response

please find the attached file

Reviewer 2 Report (New Reviewer)

This work investigates the influence of cracks of conductive textronic layers from repeated bending on their conductive properties by the numerical simulations. Comparing the simulation results with the experimental results, a relationship was derived that allows to link the number of iterations of the procedure with the number of bends of the textronic conductive layer. The manuscript is well written and data presented is reasonable, so I think it can be published after minor revisions as below:

1.     In the first few paragraphs of Introduction, the authors introduce a lot of things including wearable electronics and sensors. Please add some related references here, such as “Kim J, et al. Wearable biosensors for healthcare monitoring[J]. Nature biotechnology, 2019, 37(4): 389-406.”, “Hong Y, et al. Highly anisotropic and flexible piezoceramic kirigami for preventing joint disorders[J]. Science Advances, 2021, 7(11): eabf0795.” and “Fan W, et al. Machine-knitted washable sensor array textile for precise epidermal physiological signal monitoring[J]. Science advances, 2020, 6(11): eaay2840.”

2.     In page 4, the author claims “The thickness of the conductive layers obtained in the PVD process does not exceed 1 micrometer, which, compared to their other dimensions (in the order of millimeters), justifies treating them as infinitely thin.” Beside the thickness of the conductive layers, the thickness of substrate should be considered because the substrate thickness significantly influences the stress distribution of the conductive layers even under the same bending angle.

Author Response

please find the attached file

Reviewer 3 Report (New Reviewer)

1 English language needs to be improved. Some sentences in the manuscript are confusing.

For example, In the abstract, “The influence of defects resulting from repeated bending of the conductive layer on its conductivity, taking into account their anisotropic and partly stochastic character is analyzed.”

“This allowed to derive a formula linking the iteration number of the simulation procedure with the number of bends.”

 2 In the introduction section, there are a lot of descriptions about the importance of wearable electronics and the methods of preparing wearable electronics. However, the work turns out to be a simulation study instead of an experiment study. The authors didn’t review any existing simulation methods applied on similar topics, making it hard to assess the novelty and contribution of this study.

 3 The figures in the manuscript are not clear. The ruler in Figure 1 is not clear. The text information in Figures 9,10 and 11 is not clear.

 4 As shown in section 3.2, the authors suggested that there was a good agreement between their simulation results and the experiment results. However, they didn't compare their method with other existing methods, nor did they discuss the significance of their findings. If the findings indeed deepen the understanding of any phenomenon happening during the repetitive bending, what insights can be derived? How can the manufacturers make use of the conclusions of this work?

Round 2

Reviewer 1 Report (Previous Reviewer 2)

Dear authors

Thanks for the explanation given in your reply. Considering the arguments therein provided, it would be adequate to include them in the revised version of the manuscript; this may help future readers to understand why quantum tunneling (QT) was not considered in your simulation.

Similarly, when percolation paths are partially destroyed (due to repeated bending), isn’t quantization of resistance expected to occur along the percolation path? [Ref 1]

Ref 1: doi: https://doi.org/10.1016/0921-4526(95)00602-8

Author Response

Dear reviewer,

Attached please find the improved and upgraded text of our article.

Reviewer 3 Report (New Reviewer)

I suggest that the current manuscript deserves publication Sensors

Author Response

Dear reviewer,

Thank you very much for the positive feedback.

Please find the improved and upgraded text of our article.

This manuscript is a resubmission of an earlier submission. The following is a list of the peer review reports and author responses from that submission.

Round 1

Reviewer 1 Report

1- The study is in a valuable field in terms of the subject, but the scientific contribution it provides is almost non-existent. It is obvious that any defect will create a valley or an insulator. It is certain that this situation will change the conductivity. There is no such thing as defect detection, segmentation or testing. In other words, this modeling study is a declaration of the known.

2- The problem and scientific contribution were not adequately explained in the literature review. The problem should be clearly presented with up-to-date sources and the contributions of the proposed method should be listed as items.

3- The fact that the finite element analysis for the flow is 2D has, in my opinion, caused the study to remain sterile. The resulting defects are not actually 2D. 3D evaluation and modeling is required. What are the limitations, assumptions, assumptions and conditions of the proposed method? What advantages do they have over existing methods? or do you have?

4- The code, simulation, data etc. of the proposed method should be shared with the reviewers and editors. Otherwise, the paper should be rejected due to suspicion of plagiarism or fabrication.

5- There is no comparison/benchmarking with traditional and current methods, so the discussion part is insufficient.

Reviewer 2 Report

Dear authors

I have overall enjoyed manuscript reading. The topic discussed by the authors is interested to the audience and in general the manuscript is well written. I also recognize the originality in discussing about micro cracks in flexible electronics; most articles report about flexible sensor manufacturing and characterization, but rarely discuss about practical implications such as micro cracks.  I list below some major and minor changes that must be addressed before further article processing:

Major changes

Section 2.1: Given the statements 1) to 6) from this section, I must disagree in that quantum effects were not taken into account in the simulation. Among others, thin film nanocomposites typically exhibit: quantum tunneling, quantization of resistance and anisotropic conductivity (especially when dealing with carbon nanotubes). Is there a specific reason to ignore quantum effects in the simulation?

Figures 3 through 5 show the simulation results under multiple test conditions. However, there is some missing information in these plots; they are next listed:

A)     What are dimensions of the x and y axes? Similarly, what is the depth of the material (not shown in the plot)?

B)      Figures 4 and 5 need a color map, i.e. what is the scale of J?

C)      All the plots reported a 2-D analysis, what is the expected output for a 3-D simulation?

D)     Some data for the simulation are missing, e.g. what is the triangle dimension considered for the finite element analysis?

What is the origin of Figure 1? The authors used this figure to validate the simulation results, but did not provide a description for the figure’s origin.

Finally, I leave to the editor the decision to consider the acceptance of a manuscript that is entirely supported in simulation results. I am not aware whether this journal accepts simulation-only papers.

Minor changes:

The statement in page 2 “The PVD technique uses various …or mixed” needs a citation to support its validity.